# Association between Parity and Preterm Birth—Retrospective Analysis from a Single Center in Poland

**DOI:** 10.3390/healthcare11121763

**Published:** 2023-06-15

**Authors:** Monika Szyszka, Ewa Rzońca, Sylwia Rychlewicz, Grażyna Bączek, Daniel Ślęzak, Patryk Rzońca

**Affiliations:** 1Department of Human Anatomy, Faculty of Health Sciences, Medical University of Warsaw, 02-004 Warsaw, Poland; monika.szyszka@wum.edu.pl; 2Department of Obstetrics and Gynecology Didactics, Faculty of Health Sciences, Medical University of Warsaw, 00-575 Warsaw, Poland; ewa.rzonca@wum.edu.pl (E.R.); grazyna.baczek@wum.edu.pl (G.B.); 3St. Sophia’s Specialist Hospital, Żelazna Medical Center, 01-004 Warsaw, Poland; s.rychlewicz@szpitalzelazna.pl; 4Department of Medical Rescue, Medical University of Gdańsk, 80-210 Gdańsk, Poland; daniel.slezak@gumed.edu.pl

**Keywords:** preterm birth, primipara, multipara

## Abstract

Preterm births and parity are two medical areas that seem to be entirely different from each other. The aim of this study was to analyze the relationships between parity and maternal and neonatal outcomes associated with preterm birth. This study involved a retrospective analysis of electronic medical records from St. Sophia Hospital in Warsaw (Poland). This study was conducted among women who gave birth to preterm infants between 1 January 2017 and 31 December 2021. A total of 2043 cases of preterm births were included in the final analysis. A higher odds ratio of preterm birth in primiparas was found in women living in a city/town (OR = 1.56) and having secondary (OR = 1.46) and higher education (OR = 1.82). Multiparas who gave birth to preterm infants were more frequently diagnosed with gestational diabetes (19.69%) than primiparas. Multiparas were more likely to give birth to preterm infants who received an Apgar score of ≤7 both at 1 and 5 min after birth (25.80% and 15.34%). The results of our study emphasize the differences between primiparas and multiparas who give birth to preterm infants. Knowledge of these differences is essential to improve the perinatal care provided to mothers and their infants.

## 1. Introduction

According to the definition by the World Health Organization (WHO), preterm birth occurs before 37 weeks of pregnancy are completed. Differentiation between miscarriage and preterm birth depends on the country. In Poland, the cut-off is 22 completed weeks of pregnancy [1].

Preterm births are a global problem entailing a number of consequences for the newborn child, their family and the healthcare system [2,3]. Prematurity, the predominant complication of preterm birth, is associated with high perinatal morbidity and mortality of newborns [4,5,6]. It needs to be emphasized that during the adaptation period, preterm infants are at risk of various problems such as thermoregulatory disorders, periventricular-intraventricular hemorrhage, hyperbilirubinemia, infections, respiratory distress syndrome, necrotizing enterocolitis and retinopathy of prematurity [7]. In turn, late complications linked to premature birth include the risk of asthma, obesity, hypertension and chronic kidney disease [8]. This is a major challenge for all preterm babies’ caregivers, who need specialist equipment, training and access to current scientific reports in this field [9,10,11,12]. Therefore, preterm births are currently the subject of numerous studies and an important area of focus for many researchers, especially in the context of risk factor identification [13,14,15,16,17,18].

Globally, the number of preterm births decreased by 5.26%, from 16.06 million in 1990 down to 15.22 million in 2019. The number of deaths in this group decreased by 47.71%, from 1.27 million in 1990 down to 0.66 million in 2019 [19]. In 2019, 1 in 10 infants were born prematurely in the United States [20]. In Europe, the preterm birth rate ranges between 5% and 10% [21]. In Poland, about 8% of all births in 2017 were preterm births [22].

The multitude of risk factors for preterm birth makes it difficult to precisely predict its occurrence. However, having knowledge of these factors allows for the provision of personalized and professional care to the pregnant woman, thus reducing the number of complications associated with preterm birth [23]. These medical and non-medical factors include preterm birth in medical history, multiple pregnancies, chronic hypertension, diabetes, genetic factors, stress, mental disorders and lifestyle [18,19,20,21,22,23]. Importantly, the risk of preterm birth is also associated with parity [24]. A primipara is a woman who has given birth for the first time, whereas a multipara is a woman who has given birth more than one time. It is known that parity is a factor associated with various problems affecting women in the perinatal period [25,26,27].

However, it has been observed that the subject of preterm births has not been extensively analyzed in the context of parity [24]. Therefore, we decided to conduct our own research in this area. The aim of this study was to analyze the relationships between parity (primiparas vs. multiparas) and maternal and neonatal outcomes associated with preterm birth.

## 2. Materials and Methods

### 2.1. Study Design and Setting

This study involved a retrospective analysis of electronic medical records from St. Sophia Hospital in Warsaw (Poland), which is a tertiary referral hospital competent and licensed to manage the most difficult cases in obstetrics, gynecology and neonatal pathology, including preterm infants with extremely low birth weight. The study was conducted among women who gave birth to preterm infants between 1 January 2017 and 31 December 2021.

### 2.2. Eligibility Criteria

The study included all women who gave birth to preterm infants (this includes both spontaneous preterm births and indicated inductions for maternal/fetal indications), i.e., between 22 and 37 weeks of pregnancy, with a division into groups by parity—control group (primiparas) and study group (multiparas). The exclusion criteria were gaps in medical records and birth after 37 weeks of pregnancy. Medical records comprising 32,937 births given in the study period were analyzed. Of these, based on the eligibility criteria, 2043 cases of preterm births were included in the final analysis. The numbers of primiparas and multiparas were 1078 and 965, respectively (Figure 1).

### 2.3. Data Collection

The electronic databases obtained provided the following information: patient age, place of residence, marital status, education, obstetric history, course of pregnancy and delivery and birth data of the infant. The data were collected on the basis of the patient’s medical records from the course of the current pregnancy and the gynecological and obstetric interview as well as the general interview. The data were collected by the hospital staff, i.e., gynecologists and midwives.

### 2.4. Ethics

The study design was submitted to and approved by the Bioethical Committee at the Medical University of Warsaw (AKBE/112/2022). The study was conducted in line with the principles set out in the Declaration of Helsinki (1964) as amended. Database reports were anonymized and did not allow for patient identification at any stage of the study.

### 2.5. Statistical Analysis

The obtained data were analyzed using IBM SPSS Statistics 25.0 for Windows (Armonk, NY: IBM Corp.). Categorical variables were reported as numbers (n) and percentages (%), whereas continuous variables were reported as medians (Me) and interquartile ranges (IQR). The normality of distribution of the continuous variables was verified using the Kolmogorov–Smirnov test and the Lilliefors test. To compare the baseline data, a chi-square test was used for categorical variables and the Mann–Whitney U test for continuous variables. The odds of a given outcome occurring in the study group vs. the control group were calculated as an odds ratio (OR) with a 95% confidence interval (CI 95%). The *p*-value of <0.05 was considered statistically significant.

## 3. Results

Multiparas who gave birth to preterm infants were older than primiparas (34 vs. 31 years). Multiparas were more likely to live in rural areas (25.70%), have secondary (19.38%) or primary education (5.70%) and be in a relationship (82.38%). A higher odds ratio of preterm birth in primiparas was found in women living in a city/town (OR = 1.56) and having higher education (OR = 1.82), whereas a lower odds ratio was observed in married women. These relationships were statistically significant (*p* < 0.05). Detailed data are presented in Table 1.

The analysis showed that multiparas were significantly more likely to be pregnant for the third time (3 vs. 1) and have a history of miscarriage (35.44%) as compared to primiparas (*p* < 0.05). In the primipara group, the odds ratio was higher for twin pregnancy (OR = 1.39) and anti-biotic prophylaxis (OR = 1.35)—Table 2.

Multiparas who gave birth to preterm infants were more frequently diagnosed with gestational diabetes (19.69%) than primiparas. Primiparas, on the other hand, were more frequently diagnosed with hypertension (10.20%), pre-eclampsia (10.20%), cholestasis of pregnancy (6.59%) and hypothyroidism (27.46%) as compared to multiparas (*p* < 0.05), which is also reflected by the higher odds ratio. Moreover, in the primipara group, the odds ratio was higher for health problems (OR = 1.45). Detailed data are presented in Table 3.

Table 4 presents an analysis of the relationships between parity and selected labor variables. The analysis demonstrated that labor stimulation (8.44%), oxytocin administration in the 1st (9.37%) and 2nd (11.04%) stages of labor, epidural anesthesia (17.35%) and episiotomy (23.65%) were performed more often in primiparas than in multiparas, which is also reflected by the higher odds ratio. In turn, multiparas were more likely to experience perineal tears (7.98%), and the duration of their labor (225 vs. 311 min), as well as the duration of the 1st (205 vs. 293 min) and 2nd (10 vs. 25 min) stages of labor, was shorter than in primiparas. We also found a relationship between parity and the duration of a hospital stay. The observed relationships were statistically significant (*p* < 0.05).

Multiparas were more likely to give birth to preterm infants who received an Apgar score of ≤7 both at 1 and 5 min after birth (25.80% and 15.34%, respectively). In turn, primiparas were more likely to give birth to infants with lower birth weights (2300 vs. 2400 g). Preterm infants of primiparas (69.02%) had to be transferred to an NICU more often than those born to multiparas. The observed relationships were statistically significant (*p* < 0.05). Detailed data are presented in Table 5.

## 4. Discussion

Given the extensive literature on risk factors, whose management allows for decreasing the preterm birth rate, it seems valid to focus on an in-depth analysis of individual risk factors and their effect on perinatal outcomes [18,19,20,21,22,23]. Preterm births and parity are two medical areas that seem to be entirely different from each other. However, there are numerous associations between them. A better understanding of these relationships could contribute to reducing the incidence of preterm births and improving the reproductive health of women in this population [25,26].

Our study showed that multiparas who gave birth to preterm infants were older than primiparas, which has also been confirmed in studies by Koullali et al. and Alhainiah et al. [27,28]. Furthermore, Hangara and Yattinaman observed that multiparas were aged 22–27 years. This stems from the fact that women in India get married at a younger age, which is associated with earlier childbearing [29]. A study by Fuchs et al. showed that women aged 30–34 years are at lower risk of giving birth to preterm infants, whereas those aged ≥40 years are at greater risk [30].

Our study revealed that multiparas were more likely to live in rural areas and more frequently declared to have primary and secondary education, which is in line with the findings by Lei et al. [31]. The higher the education level, the lower the odds of preterm birth, with the lowest odds being observed in women with tertiary education, as demonstrated by Granes et al. [32]. The outcomes described above may also depend on procreative choices associated with career- and environment-related decisions made by women at a given age and in a given place of residence and with access to health care (especially in rural areas). Furthermore, our study demonstrated that multiparas who gave birth to premature infants were more likely to be in a relationship, which was also observed by Blitz et al. [33].

Gurung et al. found that the risk of preterm birth in primiparas in the Netherlands was higher than that observed in multiparas [34]. This is consistent with our results, which showed that over 50% of women who gave birth to preterm infants were primiparas. In contrast, a study conducted in Saudi Arabia by Alhainiah et al. found that multiparas gave birth to premature infants more often than primiparas [28]. Luo et al. observed that the odds ratio of preterm birth was higher in Chinese multiparas aged >35 years [35]. Koullali et al. conducted a study among women in the Netherlands, dividing them into those who were pregnant for the 1st, 2nd, 3rd, 4th and 5th time. This classification allowed the researchers to demonstrate that both primiparas and women who were pregnant for the 5th time were at risk for preterm delivery [27]. Our study found that multiparas who gave birth to preterm infants were more likely to be in their 3rd pregnancy.

Furthermore, we observed that primiparas who had a preterm delivery more often received antibiotic prophylaxis due to the colonization of the genital tract by Streptococcus agalactiae. This is consistent with the findings by Szylit et al., who reported that primiparas were more frequently positive for Streptococcus agalactiae [36]. 

Another aspect analyzed was the association between parity and selected health variables in the women studied. In our study in the primipara group, the odds ratio was higher for health problems, which has also been confirmed in studies by Chen et al. [37]. A study by Yong et al. showed that women who had two or more pregnancies were at higher risk for gestational diabetes [38]. Likewise, Wagan et al. found that women who had three or more children were at higher risk for gestational diabetes [39]. These findings are corroborated by our results. In our study, gestational hypertension was more often observed in primiparas. Li et al. found that women who were pregnant for the first time were at greater risk of gestational hypertension than multiparas [40]. Souter et al. demonstrated that elective labor induction at 39 weeks of gestation reduces the risk of gestational hypertension in both primiparas and multiparas [41]. In turn, Maeda et al. showed that multiparity significantly reduces the risk of pre-eclampsia [42]. In our study, primiparas were more likely to be diagnosed with pre-eclampsia and hypothyroidism as compared to multiparas. In a study by Gupta et al., multiparas were more likely to suffer from hypothyroidism, whereas hyperthyroidism was more often found in primiparas [43]. This difference may stem from different lifestyles led in Europe and Asia and from environmental factors. Toloza et al. demonstrated that subclinical hypothyroidism during pregnancy is associated with a higher risk of pre-eclampsia [44].

Our study showed that labor stimulation with oxytocin was more often used in primiparas. This is consistent with the findings by Oladapo et al., who also found that oxytocin was more often administered to women who were giving birth for the first time [45]. A study by Luo et al. found that epidural anesthesia was administered to over 50% of multiparas [46]. Our study, on the other hand, showed that epidural anesthesia was more often used in primiparas than in multiparas, which is consistent with the results obtained by Orbach–Zinger et al., [47]. These differences may stem from the quality of medical services and accessibility to medical procedures provided in various countries in the world.

Another variable analyzed was perineal trauma during labor [48,49,50,51]. Studies by Kartal et al. (term labor) and Beyene et al. (women > 28 HBD) showed that episiotomy was more frequently performed in primiparas than in multiparas [48,49], which corresponds with the results we obtained with regard to women giving birth to preterm infants. Furthermore, our study found that perineal tears were more likely to occur in multiparas than in primiparas. Wilson and Homer, on the other hand, showed that third- and fourth-degree perineal tears were more frequent in primiparas [51].

Our study demonstrated that the duration of the 1st and 2nd stages of labor and the total labor duration was longer in primiparas who gave birth to preterm infants than in multiparas, which corroborates the results obtained by Ashwal et al. in their study on women who had term delivery [52]. In turn, Rosenbloom et al. showed that the 2nd stage of labor was the longest in primiparas who received epidural anesthesia [53]. Moreover, differences in labor duration may stem from, among other factors, fitness and physical activity of the mother or medical interventions performed during labor, as emphasized by research findings [54,55,56]. At the same time, this may justify the results of our study.

Next, we analyzed the effect of preterm birth on neonatal outcomes relative to parity. Our study showed that preterm infants born to primiparas were more likely to receive an Apgar score of >7, both at 1 and 5 min after birth, had lower birth weight and more frequently required transfer to an NICU. Tadese et al. demonstrated that infants of women who had five or more children received lower Apgar scores. Lower birth weight, on the other hand, is more frequently observed in infants of mothers who had given birth to 2–4 children [57]. Kaur et al. found that the higher the number of labors, the lower the number of infants with low birth weight [58]. In contrast, Alhainiah et al. showed that transfer to an NICU was more frequently required in the case of infants born to multiparas [28].

Our study addresses two areas that are essential to the practice of obstetrics, namely parity and preterm birth, including their characteristic features, differences and potential consequences for the mother and her child.

The subject of preterm birth in association with parity is rarely described in publications, even though these two areas have much in common. Despite our greatest attention to detail, the study has some limitations. Firstly, it is retrospective in nature. The database analyzed was derived from St. Sophia Hospital’s electronic medical records, which had data gaps. The limitation of the study regarding data collection may be due to the insufficient completion of medical records by hospital staff. Secondly, despite the large sample size, our study was limited to only one facility, which does not reflect the internal standards followed in other obstetric hospitals. Thirdly, we managed to assess a number of parameters significant for the mother and her child, but due to the retrospective nature of the study, we were not able to obtain any additional data that would improve the quality of our results, e.g., indication for admission to an NICU. Therefore, it is necessary to conduct further research on preterm births and parity, as a better understanding of the associations between these two areas may contribute to a reduced risk of complications and improved care of the mother and her child.

## 5. Conclusions

Our study found that multiparas who gave birth to preterm infants tend to be older, live in rural areas, have secondary education and be in a relationship as compared to primiparas. Multiparas were more often diagnosed with gestational diabetes and primiparas with gestational hypertension, pre-eclampsia, cholestasis of pregnancy and hypothyroidism. In addition, it was observed that the odds ratio of health problems in pregnancy was higher in the primipara group.

Labor stimulation, oxytocin administration in the 1st and 2nd stages of labor, epidural anesthesia and episiotomy were performed more frequently in primiparas and perineal tears occurred more often in multiparas. Labor duration was shorter in multiparas than in primiparas.

Multiparas were more likely to give birth to preterm infants who received an Apgar score of ≤7 both at 1 and 5 min after birth. In turn, primiparas were more likely to give birth to infants with lower birth weights.

The results of our study emphasize the differences between primiparas and multiparas who give birth to preterm infants. Knowledge of these differences is essential to improve the perinatal care provided to mothers and their infants.

## Figures and Tables

**Figure 1 healthcare-11-01763-f001:**
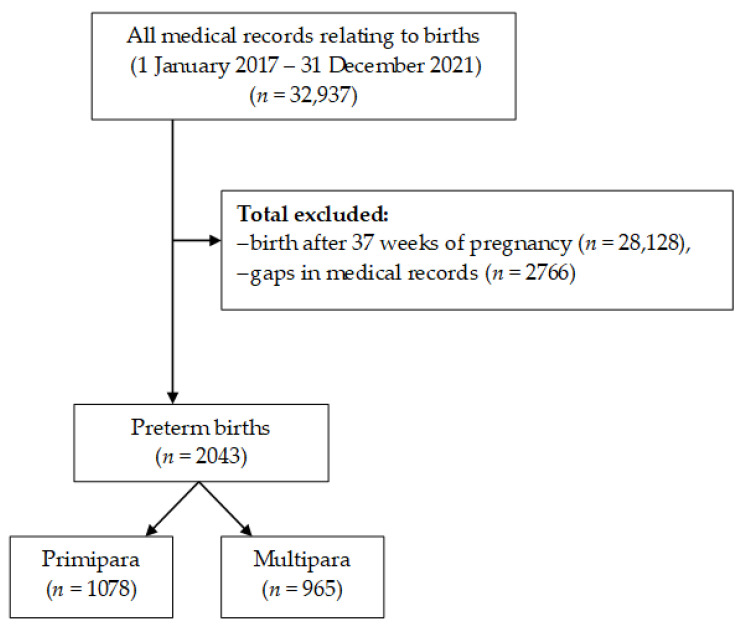
Flowchart demonstrating exclusions and final analytic sample included in this study.

**Table 1 healthcare-11-01763-t001:** Analysis of relationships between parity and selected demographic variables.

Variables	Total *n* = 2043	Primipara *n* = 1078 (52.77%)	Multipara *n* = 965 (47.23%)	OR (95% CI)	*p*-Value
Age—Me (IQR)	33 (29–36)	31 (28–34)	34 (31–37)	-	<0.01
Place of residence—*n* (%)					
Village	444 (21.83)	196 (18.18)	248 (25.70)	1	<0.01
City/town	1599 (78.17)	882 (81.82)	717 (74.30)	1.56 (1.26–1.92)	
Education—*n* (%)					
Primary education	91 (4.45)	36 (3.34)	55 (5.70)	1	
Secondary education	366 (17.91)	179 (16.60)	187 (19.38)	1.46 (0.92–2.33)	0.111
Higher education	1586 (77.63)	863 (80.06)	723 (74.92)	1.82 (1.18–2.81)	0.006
Marital status—*n* (%)					
Single	450 (22.03)	280 (25.97)	170 (17.62)	1	<0.01
In a relationship	1593 (77.97)	798 (74.03)	795 (82.38)	0.61 (0.49–0.76)	
COVID-19 Era—*n* (%)					
No	1432 (70.09)	763 (70.78)	669 (69.33)	1	0.474
Yes	611 (29.91)	315 (29.22)	296 (30.87)	0.93 (0.77–1.13)	

OR—odds ratio, 95% CI—95% confidence interval, IQR—interquartile range.

**Table 2 healthcare-11-01763-t002:** Analysis of relationships between parity and selected obstetric variables.

Variables	Total	Primipara *n* = 1078	Multipara *n* = 965	OR (95% CI)	*p*-Value
No. of pregnancies—Me (IQR)	2 (1–3)	1 (1–1)	3 (2–3)	-	<0.01
HBD—Me (IQR)	35 (32–36)	35 (33–36)	35 (32–36)	-	0.341
Pregnancy type—*n* (%)					
Single	1527 (74.74)	777 (72.08)	750 (77.72)	1	
Twin	474 (23.20)	280 (25.97)	194 (20.10)	1.39 (1.13–1.72)	0.002
Triplet	42 (2.06)	21 (1.95)	21 (2.18)	0.97 (0.52–1.78)	0.910
History of miscarriage—*n* (%)					
No	1492 (73.03)	869 (80.61)	623 (64.56)	1	<0.01
Yes	551 (26.97)	209 (19.39)	342 (35.44)	0.44 (0.36–0.64)	
Pessary—*n* (%)					
No	1848 (90.46)	963 (89.33)	885 (91.71)	1	0.068
Yes	195 (9.54)	115 (10.67)	80 (8.29)	1.32 (0.98–1.78)	
GBS—*n* (%)					
No	1477 (72.30)	781 (72.45)	696 (72.12)	1	0.369
Yes	169 (8.27)	85 (7.88)	84 (8.70)	0.90 (0.66–1.24)	
No	397 (19.43)	212 (19.67)	185 (19.17)	1.02 (0.82–1.28)	
Thromboprophylaxis—*n* (%)					
No	809 (39.60)	432 (40.07)	377 (39.07)	1	0.642
Yes	1234 (60.40)	646 (59.93)	588 (60.93)	0.96 (0.80–1.15)	
Antibiotic prophylaxis—*n* (%)					
No	266 (13.02)	123 (11.41)	143 (14.82)	1	0.022
Yes	1777 (86.98)	955 (88.59)	822 (85.18)	1.35 (1.04–1.57)	

OR—odds ratio, 95% CI—95% confidence interval, IQR—interquartile range, GBS—Streptococcus agalactiae.

**Table 3 healthcare-11-01763-t003:** Analysis of relationships between parity and selected health problems.

Variables	Total	Primipara *n* = 1078	Multipara *n* = 965	OR (95% CI)	*p*-Value
Gestational diabetes—*n* (%)					
No	1680 (82.23)	905 (83.95)	775 (80.31)	1	0.032
Yes	363 (17.77)	173 (16.05)	190 (19.69)	0.78 (0.62–0.98)	
Gestational hypertension—*n* (%)					
No	1866 (91.34)	968 (89.90)	898 (93.06)	1	0.009
Yes	177 (8.66)	110 (10.20)	67 (6.94)	1.52 (1.11–2.09)	
Pre-eclampsia—*n* (%)					
No	1877 (91.87)	968 (89.80)	909 (94.20)	1	<0.01
Yes	166 (8.13)	110 (10.20)	56 (5.80)	1.85 (1.32–2.58)	
Cholestasis of pregnancy—*n* (%)					
No	1936 (94.76)	1007 (93.41)	929 (96.27)	1	0.004
Yes	107 (5.24)	71 (6.59)	36 (3.73)	1.82 (1.21–2.74)	
Hypothyroidism—*n* (%)					
No	1531 (74.94)	782 (72.54)	749 (77.62)	1	0.008
Yes	512 (25.06)	296 (27.46)	216 (22.38)	1.31 (1.07–1.61)	
Hashimoto’s—*n* (%)					
No	1893 (82.66)	1001 (92.86)	892 (92.44)	1	0.715
Yes	150 (7.34)	77 (7.14)	73 (7.56)	0.94 (0.67–1.31)	
Anemia—*n* (%)					
No	1014 (49.63)	516 (47.87)	498 (51.61)	1	0.091
Yes	1029 (50.37)	562 (52.13)	467 (48.39)	0.86 (0.72–1.02)	
Thrombocytopenia—*n* (%)					
No	1787 (87.47)	934 (86.64)	853 (88.39)	1	0.233
Yes	256 (15.53)	144 (13.36)	112 (11.61)	1.17 (0.90–1.53)	
Cervical incompetence—*n* (%)					
No	1982 (97.01)	1039 (96.38)	943 (97.72)	1	0.076
Yes	61 (2.99)	39 (3.62)	22 (2.28)	1.61 (0.85–2.73)	
Health Problems—*n* (%)					
No	384 (18.80)	174 (16.14)	210 (21.76)	1	0.001
Yes	1659 (81.20)	904 (83.86)	755 (78.24)	1.45 (1.16–1.81)	

**Table 4 healthcare-11-01763-t004:** Analysis of relationships between parity and selected delivery variables.

Variables	Total	Primipara *n* = 1078	Multipara *n* = 965	OR (95% CI)	*p*-Value
Labor type—*n* (%)					
Physiologic	748 (36.61)	388 (35.99)	360 (37.31)	1	
C-section	1281 (62.70)	680 (63.08)	601 (62.28)	1.05 (0.88–1.26)	0.598
Intervention	14 (0.69)	10 (0.93)	4 (0.41)	2.32 (0.72–7.46)	0.158
Family member present—*n* (%)					
No	1625 (79.54)	851 (78.94)	774 (80.21)	1	0.479
Yes	191 (20.46)	227 (21.06)	191 (19.79)	1.08 (0.87–1.34)	
Pre-induction—*n* (%)					
No	2008 (98.29)	1058 (98.14)	950 (98.45)	1	0.601
Yes	35 (1.71)	20 (1.86)	15 (1.55)	1.20 (0.61–2.35)	
Induction—*n* (%)					
No	1867 (91.39)	976 (90.54)	891 (92.33)	1	0.149
Yes	176 (8.61)	102 (9.46)	74 (7.67)	1.26 (0.92–1.72)	
Stimulation—*n* (%)					
No	1917 (93.83)	987 (91.56)	930 (96.37)	1	<0.01
Yes	126 (6.17)	91 (8.44)	35 (3.63)	2.45 (1.64–3.66)	
Oxytocin—stage 1 *—*n* (%)					
No	1888 (92.41)	977 (90.63)	911 (94.40)	1	0.001
Yes	155 (7.59)	101 (9.37)	54 (5.60)	1.74 (1.24–2.46)	
Oxytocin—stage 2 **—*n* (%)					
No	1859 (90.99)	989 (88.96)	900 (93.26)	1	0.001
Yes	184 (9.01)	119 (11.04)	65 (6.74)	1.72 (1.25–2.36)	
Oxytocin—stage 3 ***—*n* (%)					
No	1529 (74.84)	797 (73.93)	732 (75.85)	1	0.318
Yes	514 (25.16)	281 (26.07)	233 (24.15)	1.11 (0.91–1.35)	
Amniotomy—*n* (%)					
No	2024 (99.07)	1068 (99.07)	956 (99.07)	1	0.991
Yes	19 (0.93)	10 (0.93)	9 (0.97)	0.99 (0.40–2.46)	
Epidural anesthesia—*n* (%)					
No	1761 (86.20)	891 (82.65)	870 (90.16)	1	<0.01
Yes	282 (13.80)	187 (17.35)	95 (9.84)	1.92 (1.48–2.50)	
Perineal trauma—*n* (%)					
No	1559 (76.31)	783 (72.63)	776 (80.41)	1	
Perineal tear	117 (5.73)	40 (3.71)	77 (7.98)	0.52 (0,35–0,76)	0.001
Episiotomy	367 (17.96)	255 (23.65)	112 (11.61)	2.26 (1.77–2.88)	<0.01
Uterine curettage—*n* (%)					
No	1724 (84.39)	899 (83.40)	825 (85.49)	1	0.192
Yes	319 (15.61)	179 (16.60)	140 (14.51)	1.17 (0.92–1.49)	
Labor duration—stage 1 (min)—Me (IQR)	240 (170–360)	293 (200–405)	205 (150–290)	-	<0.01
Labor duration—stage 2 (min)—Me (IQR)	16 (10–30)	25 (15–40)	10 (8–20)	-	<0.01
Labor duration—stage 3 (min)—Me (IQR)	10 (10–10)	10 (10–10)	10 (10–10)	-	0.432
Labor duration (min)—Me (IQR)	280 (195–400)	331 (240–445)	225 (170–313)	-	<0.01
Blood loss (ml)—Me (IQR)	500 (400–500)	500 (400–500)	500 (350–500)	-	0.590
Hospital stay (days)—Me (IQR)	8 (6–13)	8 (6–13)	8 (5–13)	-	<0.01

*—to induction of labor and stimulate contraction of the uterus; **—to stimulate contraction of the uterus; ***—active management reduces mean maternal blood loss.

**Table 5 healthcare-11-01763-t005:** Analysis of relationships between parity and selected neonatal variables.

Variables	Total	Primipara *n* = 1078	Multipara *n* = 965	OR (95% CI)	*p*-Value
1-min APGAR score—*n* (%)					
≤7	479 (23.45)	230 (21.34)	249 (25.80)	1	0.017
>7	1564 (76.55)	848 (78.66)	716 (74.20)	1.28 (1.05–1.57)	
5-min APGAR score—*n* (%)					
≤7	280 (13.71)	132 (12.24)	148 (15.34)	1	0.043
>7	1763 (86.29)	946 (87.76)	817 (84.66)	1.30 (1.01–1.67)	
Birth weight (grams)—Me (IQR)	2340 (1750–2750)	2300 (1750–2690)	2400 (1730–2800)	-	0.018
NICU transfer—*n* (%)					
No	676 (33.09)	334 (30.98)	342 (35.44)	1	0.033
Yes	1367 (66.91)	744 (69.02)	623 (64.56)	1.22 (1.02–1.47)	

## Data Availability

The data presented in this study are available on request from the corresponding author.

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
