# Peer review of "Association between Parity and Preterm Birth—Retrospective Analysis from a Single Center in Poland"

_healthcare, 2023, doi:10.3390/healthcare11121763_

Round 1

Reviewer 1 Report

First of all, I would like to congratulate and thank you for the work done in this scientific article.

I would like to make several comments regarding your manuscript:

1. How were the data collected? How many individuals collected the data? Is there any unidentified bias in the limitations of the study regarding data collection?

2. In Table 1, you need to make the following modifications:

a. Unify the criteria for using "n" in uppercase or lowercase.

b. There is no value of p 0.000, indicate if necessary p<0.01, but not an absolute value of 0.

c. The OR for secondary education is 1.46, however, the CI (0.92-2.33) does not support the conclusion stated in lines 107-110.

3. In Table 2, you need to make the following modifications:

a. In the section on type of pregnancy, there is an OR of 0.97 for triplet pregnancies; however, the CI (0.52-1.78) does not support the conclusion mentioned in line 115.

b. What criteria did you use to categorize patients in the diagnosis of gestational hypertension and pre-eclampsia?

c. There is no value of p 0.000, indicate if necessary p<0.01, but not an absolute value of 0.

4. In Table 3, you need to make the following modifications:

a. There is no value of p 0.000, indicate if necessary p<0.01, but not an absolute value of 0.

5. In Table 4, you need to provide the following clarifications:

a. There is no value of p 0.000, indicate if necessary p<0.01, but not an absolute value of 0.

b. Specify what each stage of labor in oxytocin stimulation consists of.

c. Have you collected and analyzed data on perineal trauma in primiparous and multiparous women who did not have an episiotomy? It is logical that women who have had an episiotomy would not have a tear, so the conclusion that multiparous women are at a higher risk of perineal trauma may not be accurate if this confounding factor is not controlled. If you have not done so, please consider this variable and analyze perineal trauma between multiparous and primiparous women who did not have an episiotomy. Similarly, analyze the differences between multiparous and primiparous women in terms of an intact perineum, i.e., without episiotomy or tear. These results and conclusions will be more reliable.

6. Rewrite the discussion section, lines 211-217 may change depending on the changes in the results.

7. Rewrite the conclusions if necessary, as they may be modified based on the suggested analysis of perineal injury.

8. Indicate un the title the region or country of the retrospective analysis. 

Author Response

Dear Reviewer,

We are very grateful for the time you have taken to review our paper and for all of your suggestions.

Comments and Suggestions for Authors

First of all, I would like to congratulate and thank you for the work done in this scientific article.

I would like to make several comments regarding your manuscript:

  1. How were the data collected? How many individuals collected the data? Is there any unidentified bias in the limitations of the study regarding data collection?

Thank you for your remark. We have expanded the description of the material and methods and added limitations about the analysis of electronic medical documentation in the discussion.

  1. In Table 1, you need to make the following modifications:
  2. Unify the criteria for using "n" in uppercase or lowercase.
  3. There is no value of p 0.000, indicate if necessary p<0.01, but not an absolute value of 0.
  4. The OR for secondary education is 1.46, however, the CI (0.92-2.33) does not support the conclusion stated in lines 107-110.

Following the Reviewer’s suggestion, we have restructured the table 1 and re-described the results.

  1. In Table 2, you need to make the following modifications:
  2. In the section on type of pregnancy, there is an OR of 0.97 for triplet pregnancies; however, the CI (0.52-1.78) does not support the conclusion mentioned in line 115.
  3. What criteria did you use to categorize patients in the diagnosis of gestational hypertension and pre-eclampsia?

Thank you for your remark. We would like to emphasize that we relied on electronic medical documentation, which is completed by the doctor during the admission of a pregnant woman / giving birth to the hospital. The gynecologist makes the diagnosis based on the current recommendations of the Polish Society of Gynecologists and Obstetricians, and this was the basis for our analysis.

  1. There is no value of p 0.000, indicate if necessary p<0.01, but not an absolute value of 0.

It has been modified as suggested by the Reviewer.

  1. In Table 3, you need to make the following modifications:
    1. There is no value of p 0.000, indicate if necessary p<0.01, but not an absolute value of 0.

It has been modified as suggested by the Reviewer.

  1. In Table 4, you need to provide the following clarifications:
  2. There is no value of p 0.000, indicate if necessary p<0.01, but not an absolute value of 0.
  3. Specify what each stage of labor in oxytocin stimulation consists of.

It has been modified as suggested by the Reviewer.

  1. Have you collected and analyzed data on perineal trauma in primiparous and multiparous women who did not have an episiotomy? It is logical that women who have had an episiotomy would not have a tear, so the conclusion that multiparous women are at a higher risk of perineal trauma may not be accurate if this confounding factor is not controlled. If you have not done so, please consider this variable and analyze perineal trauma between multiparous and primiparous women who did not have an episiotomy. Similarly, analyze the differences between multiparous and primiparous women in terms of an intact perineum, i.e., without episiotomy or tear. These results and conclusions will be more reliable.

Following the Reviewer’s suggestion, we have reanalyzed this aspect and re-described table 4 and  the results. We hope that this form of presentation will be satisfactory.

  1. Rewrite the discussion section, lines 211-217 may change depending on the changes in the results.

As suggested, we reviewed the perineal trauma discussion, but new analysis did not change previous results and conclusions.

  1. Rewrite the conclusions, if necessary, as they may be modified based on the suggested analysis of perineal injury.

As suggested, we verified the conclusions as per Reviewer comments, which did not change the results.

  1. Indicate un the title the region or country of the retrospective analysis.

Following the Reviewer’s suggestion, we have changed the title - Association between Parity and Preterm Birth - Retrospective Analysis from a Single Center in Poland

All the remarks and suggestions addressed in the text have been marked in red.

Reviewer 2 Report

This study evaluated the associations between parity and preterm birth.  My greatest concern with this study is in the materials and methods section of the study. You have included all women who have preterm birth. This included not only those who come in preterm labor and deliver prematurely  but also those women with hypertension, preterm premature rupture of the membranes, pregnancies complicated fetal growth restriction and others who are delivered preterm because of a maternal and/or fetal indication.  There is a significant difference between women who spontaneously deliver preterm and those that are delivered intentionally early because of maternal/fetal concerns. What the health care provider ultimately want to know is if I had a preterm delivery is it going to happen again. So the question to be answered should be what is the relationship of parity to who spontaneously deliver preterm and what is the relationship to women delivered preterm because of maternal/fetal concerns. 

Author Response

Dear Reviewer,

We are very grateful for the time you have taken to review our paper and value your remarks.

Comments and Suggestions for Authors

This study evaluated the associations between parity and preterm birth.  My greatest concern with this study is in the materials and methods section of the study. You have included all women who have preterm birth. This included not only those who come in preterm labor and deliver prematurely  but also those women with hypertension, preterm premature rupture of the membranes, pregnancies complicated fetal growth restriction and others who are delivered preterm because of a maternal and/or fetal indication.  There is a significant difference between women who spontaneously deliver preterm and those that are delivered intentionally early because of maternal/fetal concerns. What the health care provider ultimately want to know is if I had a preterm delivery is it going to happen again. So the question to be answered should be what is the relationship of parity to who spontaneously deliver preterm and what is the relationship to women delivered preterm because of maternal/fetal concerns.

Thank you for the important comment regarding indications for preterm delivery. In response to the Reviewer's comment, we have included an analysis of the relationship between maternal indications and parity. However, we would like to emphasize that these are selected indications. It is important to note that the objective of our study was not to assess the impact of maternal/fetal indications on the occurrence of preterm delivery in the context of parity, but rather to present the characteristics of preterm birth and maternal and neonatal outcomes in relation to preterm deliveries and parity. The valuable comment and suggestion from the Reviewer require the planning and conduct of completely new studies, which will be based on the analysis of maternal/fetal indications in order to address the specific question raised in the review, which will be the subject of future research.

All the remarks and suggestions addressed in the text have been marked in red.

Round 2

Reviewer 1 Report

Thank you very much for your corrections and congratulations for your work.

Author Response

Dear Reviewer,

We are very grateful for the time you have taken to rewiev our paper. We are glad that the changes we have made improve the overall quality of the our manuscript in the line with your expectations. 

Reviewer 2 Report

Line 75 would add the statement after who gave birth to preterm infants " (this includes both spontaneous preterm births and indicated inductions for maternal/fetal indications)"

Author Response

Dear Reviewer,

We are very grateful for the time you have taken to rewiev our paper. 
Following the suggestion, we have added the information in line 75. 
We hope that the changes we have made improve the overall quality of the our manuscript in the line with your expectations.